# Effect of Surfactants on the Binding Properties of a Molecularly Imprinted Polymer

**DOI:** 10.3390/polym14235210

**Published:** 2022-11-30

**Authors:** Valentina Testa, Laura Anfossi, Simone Cavalera, Matteo Chiarello, Fabio Di Nardo, Thea Serra, Claudio Baggiani

**Affiliations:** Department of Chemistry, University of Torino, Via Giuria 7, 10125 Torino, Italy

**Keywords:** molecularly imprinted polymer, 2,4,5-T, surfactant, Tween 20, SDS, CTAB, binding affinity, binding selectivity, imprinting effect

## Abstract

In molecularly imprinted polymers, non-specific interactions are generally based on weak forces between the polymer surface and the sample matrix. Thus, additives able to interfere with such interactions should be able to significantly reduce any non-specific binding effect. Surfactants represent an interesting class of substances as they are cheap and easily available. Here, we present a study of the effect of three surfactants (the anionic sodium dodecylsulphate, SDS, the cationic cetyltrimethylammonium bromide (CTAB) and the non-ionic polyoxyethylene-(20)-sorbitan monolaurate Tween 20) on the binding affinity of a 2,4,5-trichlorophenoxyacetic acid (2,4,5-T)-imprinted polymer for the template and its analogue 2,4-dichlorophenoxyacetic acid (2,4-D). The experimental results indicate that increasing amounts of surfactant decrease the binding affinity for the ligands strongly for the ionic ones, and more weakly for the non-ionic one. This effect is general, as it occurs for both 2,4,5-T and 2,4-D and for both the imprinted and the not-imprinted polymers. It also proves that the magnitude of this effect mainly depends on the presence or absence of an ionic charge, and that the hydrophobic “tail” of surfactants plays only a minor role.

## 1. Introduction

The binding selectivity typical of molecularly imprinted polymers (MIPs) is of fundamental importance in many analytical applications, such as solid phase extraction [1,2,3,4], immunochemical assays [5,6,7] and sensoristics [8,9,10,11]. In order to preserve selectivity, it is important to ensure that any non-specific binding will be minimized. Since non-specific interactions are based on weak hydrophobic forces between the polymer surface and the less hydrophilic components of the sample, non-specific binding is generally attributed to hydrophobic interactions [12,13,14]. Consequently, additives interfering with such interactions should be able to significantly reduce any non-specific binding effect. Among the possible additives, surfactants represent an obvious choice, as they are cheap, easily available and compatible with the organic solvents commonly used. Despite this, in the literature the effect of these additives on the interaction between MIPs and analytes is reported only in sporadic cases [15,16,17,18], and no systematic studies have been reported to date.

The potential effects of the surfactant molecules on the binding properties of an MIP may depend on the amphipathic properties, due to the presence of a hydrophilic head (ionic or neutral) and a hydrophobic tail. In solution, these properties allow the surfactant molecules to spontaneously structure themselves in simple supramolecular aggregates, such as micelles, or even complex ones, such as double layers and lamellar or cubic phases [19]. Each surfactant is characterized by a concentration threshold (critical micellar concentration, CMC) below which supramolecular structures are no longer able to form [20]. However, even in sub-micellar conditions, the surfactants may potentially be able to establish a multiplicity of non-covalent interactions with the template molecules in solution, on the surface of the polymer and the binding sites present therein. It is therefore clear that in order to try to understand the nature of these interactions, it is opportune to examine in depth of the behavior of an MIP in the presence of different types of surfactants.

With the aim of evaluating the effect of surfactants onto the binding properties of MIPs, in this work we studied the effect of sub-micellar amounts of three different surfactants—the anionic sodium dodecylsulphate (SDS), the cationic cetyltrimethylammonium bromide (CTAB) and the non-ionic polyoxyethylene-(20)-sorbitan monolaurate (Tween 20)—in acetonitrile/water mixtures of varying composition on the binding behavior of a 2,4,5-trichlorophenoxyacetic acid-imprinted polymer chosen as system model.

## 2. Materials and Methods

### 2.1. Materials

The cetyltrimethylammonium bromide (CTAB), 2,4-dichlorophenoxyacetic acid (2,4-D), 2,2-dimethoxy-2-phenylacetophenone (DMPA), ethylene dimethacrylate, (EDMA), sodium methanesulfonate (NaMS), tetramethylammonium chloride (TMAC), 2,4,5-trichlorophenoxyacetic acid (2,4,5-T) and 4-vinylpyridine (4VP) were from Sigma–Aldrich–Fluka (Milan, Italy). The acetonitrile (MeCN, HPLC-gradient grade), acetic acid, polyethylenglycole 400 (PEG400), polyoxyethylene-(20)-sorbitan monolaurate (Tween 20) and sodium dodecylsulfate (SDS) were from VWR International (Milan, Italy).

A 2,4,5-Trichlorophenoxyacetic acid-imprinted polymer (MIP), not imprinted polymer (NIP) and ligand solutions were prepared as previously described [21]. 

### 2.2. Binding Measurements

To measure the binding isotherms in the presence of surfactants, about 40 mg of imprinted or non-imprinted polymer was exactly weighed in 3 mL flat bottom amber glass vials and 1.00 mL of acetonitrile or acetonitrile-water mixtures containing increasing amounts of phenoxyacids, ranging from 1 to 200 μg mL^−1^ and variable amounts of surfactants (Tween-20: 0–20.4 mmol L^−1^, SDS: 0–6.92 mmol L^−1^, CTAB: 0–5.49 mmol L^−1^), were added. The vials were incubated overnight at room temperature under continuous agitation on a horizontal rocking table. Then, the solutions were filtered through 0.22 μm nylon membranes, and manually transferred into 250-μL HPLC autosampler vials. The free fraction (F) of the phenoxyacid was measured by HPLC in accordance with the literature [21].

Binding parameters were calculated using SigmaPlot 12 (Systat Software Inc., Richmond, CA, USA). A non-linear least square fitting was applied to the averaged experimental data. The binding isotherm parameters were calculated using a Langmuir binding isotherm model:(1)$$B=\frac{K_{\mathrm{eq}}B_{\max}F}{1+K_{\mathrm{eq}}F}$$
where B is the phenoxyacid bound to the polymer, F is the phenoxyacid not bound, K_eq_ is the binding affinity expressed as an equilibrium binding constant and B_max_ is the binding site density. To assure robust results, weighted (1/y) Pearson VII limit minimization was chosen as the minimization method. To avoid being trapped in local minima, which would give incorrect results, minimizations were carried out several times by using different initial guess values for the binding parameters.

The imprinting factor, IF, was calculated as:(2)$$\mathrm{IF}=\frac{K_{\mathrm{eq}\left(\mathrm{MIP}\right)}}{K_{\mathrm{eq}\left(\mathrm{NIP}\right)}}$$
where K_eq(MIP)_ and K_eq(NIP)_ are the equilibrium binding constants measured on the MIP and the NIP, respectively. 

The binding selectivity, α, was calculated as:(3)$$\alpha =\frac{K_{\mathrm{eq}\left(2,4-D\right)}}{K_{\mathrm{eq}\left(2,4,5-T\right)}}$$
where K_eq(2,4-D)_ and K_eq(2,4,5-T)_ are the equilibrium binding constants calculated for 2,4-D and 2,4,5-T, respectively.

## 3. Results

In order to verify experimentally the effects of surfactants on the polymer–template binding, we measured the binding affinities (numerical values are reported in Appendix A) of the template ligand 2,4,5-T and the related 2,4-D for a fixed amount of imprinted (or not imprinted, in the case for evaluation of non-specific binding) polymer in the presence of increasing amounts of neutral, anionic or cationic surfactant. We took care to stay below the critical micellar concentration to avoid the presence of micelles in the liquid phase, so as not to overly complicate the binding equilibrium.

### 3.1. Effect on the Binding Affinity for the Template

As reported in Figure 1, Figure 2 and Figure 3, the binding affinity of 2,4,5-T for the MIP always follows the typical trend already observed previously for this type of polymer [18,19,20]: a minimum in the presence of a solution rich in acetonitrile (approx. ϕ_water_ = 0.24), followed by a progressive and significant increase as the molar fraction of water rises. 

When increasing amounts of surfactant are added to the acetonitrile–water mixture, the binding affinities invariably become lower than the correspondents measured in the absence of surfactant, with a general tendency to decrease as the concentration of surfactant increases. The depressive effect of surfactants on the binding affinity seems to be particularly marked in the case of ionic surfactants, with a prevalence of the anionic one (SDS) compared to the cationic one (CTAB). Indeed, to give some examples, in pure acetonitrile, an SDS concentration equal to 1.73 mmol L^−1^ (the lower experimentally tested) is already able to decrease the binding affinity of 2,4,5-T by approximately slightly under one order of magnitude (from 7.9 × 10^4^ L mol^−1^ to 9 × 10^3^ L mol^−1^) and confirm the decrease for the acetonitrile-water mixture 4+6 (*v/v*) (ϕ_water_ = 0.83), where the binding affinity decreases from 8.3 × 10^5^ L mol^−1^ to 1.3 × 10^5^ L mol^−1^. For CTAB, in the same conditions, the decrease is significant but less pronounced than SDS: in pure acetonitrile, a CTAB concentration equal to 1.37 mmol L^−1^ (the lower experimentally tested) decreases the binding affinity of 2,4,5-T more than three times (from 7.9 × 10^4^ L mol^−1^ to 2.1 × 10^4^ L mol^−1^), while in acetonitrile-water mixture 4+6 (*v/v*) (ϕ_water_ = 0.83), the binding affinity decreases from 8.3 × 10^5^ L mol^−1^ to 2.3 × 10^4^ L mol^−1^. On the contrary, when the neutral surfactant is used, the depressive effect is much more limited. In fact, compared to the previous example, in pure acetonitrile, a Tween 20 concentration equal to 0.81 mmol L^−1^ (also in this case the lower experimentally tested) decreases the binding affinity of 2,4,5-T only from 7.9 × 10^4^ L mol^−1^ to 7.7 × 10^4^ L mol^−1^, while in acetonitrile-water mixture 4+6 (*v/v*) (ϕ_water_ = 0.83), the binding affinity decreases from 8.3 × 10^5^ L mol^−1^ to 7.5 × 10^5^ L mol^−1^.

### 3.2. Effect on Non Specific Binding

Surfactants can potentially be able to establish a multiplicity of non-covalent interactions not only with the template molecules in solution and the binding sites present in the polymer, but also with its surface, thus interfering with the non-specific binding properties of the polymer itself. This effect can be acceptably represented by the binding of the template measured for the NIP. As reported in Figure 4a, Figure 5a and Figure 6a, in the case of NIP, the binding affinities of 2,4,5-T mirror those measured for the MIP, with the obvious difference of a lesser magnitude. In fact, the binding values show a minimum in the presence of a solution rich in acetonitrile (ϕ_water_ = 0.24) followed by a progressive and significant increase as the molar fraction of water rises. Moreover, the depressive effect of surfactants on the binding affinity seems to be particularly marked in the case of ionic surfactants, with a prevalence of the anionic one (SDS) compared to the cationic one (CTAB). In addition to this case, to give some examples, in pure acetonitrile, the three surfactants at the lowest experimentally tested concentration (Tween 20: 0.81 mmol L^−1^; SDS: 1.73 mmol L^−1^; CTAB: 1.37 mmol L^−1^) decrease the binding affinity of 2,4,5-T from 3.9 × 10^4^ L mol^−1^ to 2.1 × 10^4^ L mol^−1^ (Tween 20), 3 × 10^3^ L mol^−1^ (SDS) and 1.6 × 10^4^ L mol^−1^ (CTAB), while at the highest experimentally tested concentrations (Tween 20: 20.4 mmol L^−1^; SDS: 6.92 mmol L^−1^; CTAB: 5.49 mmol L^−1^), the binding affinity decrease only to 1.5 × 10^4^ L mol^−1^ (Tween 20), 2 × 10^3^ L mol^−1^ (SDS) and 3 × 10^3^ L mol^−1^ (CTAB). Otherwise, in acetonitrile-water mixture 4+6 (*v/v*) (ϕ_water_ = 0.83), at the lowest concentrations of surfactants, the binding affinity for 2,4,5-T decreases from 4 × 10^5^ L mol^−1^ to 2.6 × 10^5^ L mol^−1^ (Tween 20), 5.9 × 10^4^ L mol^−1^ (SDS) and 1 × 10^5^ L mol^−1^ (CTAB), while at the higher experimentally tested concentrations (Tween 20: 20.4 mmol L^−1^; SDS: 6.92 mmol L^−1^; CTAB: 5.49 mmol L^−1^), the binding affinity decrease only to 1.5 × 10^4^ L mol^−1^ (Tween 20), 2 × 10^3^ L mol^−1^ (SDS) and 3 × 10^3^ L mol^−1^ (CTAB).

The fact that both the MIP and NIP binding affinity for 2,4,5-T seems to have the same behavior in the presence of surfactants would suggest that the latter do not affect the imprinting factor. However, as reported in Figure 4b, Figure 5b and Figure 6b, small differences in binding affinity cause significant changes in IF values. In fact, in the presence of ionic surfactants, there is a weak increase, except for SDS when measured in pure acetonitrile, where IF increase markedly by about 4 times. On the contrary, Tween 20 seems to not have a significant effect on the IF values, which remain almost the same as measured in the absence of this surfactant.

### 3.3. Effect on the Binding Selectivity

Regarding the effect of surfactants on the binding selectivity, as reported in Figure 7, Figure 8 and Figure 9, the binding affinities measured for 2,4-D mirror those for 2,4,5-T: a minimum in the presence of a solution rich in acetonitrile (ϕ_water_ = 0.24), followed by a progressive and significant increase as the molar fraction of water rises. In addition, increasing amounts of surfactant exerts a depressive effect on binding affinity, confirming a general tendency to decrease as the concentration of surfactant increases. It must be noted that, despite the close similarity in the binding affinity trends measured in the presence of surfactants between 2,4,5-T and 2,4-D, small differences within the pair of equilibrium binding constant values can change the resulting binding selectivity. In fact, in the presence of Tween 20, the α-values show no significant changes compared to those measured in the absence of surfactant, while in the presence of SDS, a limited but significant increase can be observed but only for acetonitrile-rich mixtures (ϕ_water_ = 0–0.24). Lastly, contrary to what happens in the absence of surfactant, in the presence of CTAB the α-values increase progressively as the molar fraction of water increases.

## 4. Discussion

In the past, the binding properties of 2,4,5-T-imprinted polymers have been extensively studied [21]. The experimental results showed that the formation of an ion-pair between the pyridine ring in the binding site and the carboxyl group of the ligand, plus supplementary hydrophobic interactions between the binding site and the aromatic ring, is the driving force responsible for the molecular recognition effect. Therefore, in the presence of surfactant molecules capable of both polar and hydrophobic interactions, interference with the polymer–template binding in many different ways is expected, with an overall weakening of the binding interaction due to the competition between the ligand and the surfactant for the surface of the polymer and its binding sites.

The many ways a surfactant can hypothetically interact with the imprinted polymer and its template molecule are illustrated in Figure 1. Concerning the polar head of the neutral Tween 20, it can interact with the carboxyl substituent of 2,4,5-T (case “A”) and with the pyridine ring inside (case “C”) or outside (case “F”) the binding site in its indissociated forms. Secondly, the anionic SDS cannot interact with the carboxyl substituent of 2,4,5-T, but it forms an ion pair with the pyridine ring inside (case “C”) or outside (case “F”) the binding site. Lastly, the cationic CTAB cannot interact with the pyridine ring inside or outside the binding site, but it forms an ion pair with the carboxyl substituent of 2,4,5-T in its anionic form (case “A”). Concerning the hydrophobic tail, all three surfactants can interact with the aromatic ring of 2,4,5-T (case “B”), with hydrophobic structures as the polymer backbone or the aromatic ring of pyridine inside (case “D”) or outside (case “E”) the binding site. It must be noted that all six cases must be considered possible only when the polymer is imprinted but, when the polymer is not imprinted, cases C and D must be ruled out by default, and only the template-surfactant in solution (cases A and B) and the polymer surface-surfactant (cases E and F) should be considered.

The experimental results indicate that increasing amounts of surfactant have a depressing effect onto the binding affinities. This effect is general, and it occurs for all the surfactants considered and for both the imprinted and the not imprinted polymers, even if with different magnitudes. Consequently, all six types of interactions reported in Figure 1 can be considered possible. On the contrary, a hypothetical interaction possible only inside the binding sites but not outside (Figure 1, cases C & D), should manifest as a reduction of the affinity towards the ligand in the case of the MIP, but an absence of any effect in the case of the NIP. Moreover, it must be considered that the ionic surfactants, SDS and CTAB, have a much more marked effect on the binding affinity than the neutral surfactant, Tween 20. Thus, it is plausible that the magnitude of this effect mainly depends on the presence or absence of an ionic charge, and that the hydrophobic “tail” of surfactants plays only a minor role. In this case, the dominant interactions in Figure 1 would be A (ion pair in solution), C (ion pair in the binding site) and F (ion pair outside the binding site), respectively.

To demonstrate this hypothesis, we therefore measured the binding affinity between MIP and 2,4,5-T in the presence of additives similar to the polar head of surfactants: polyethylene glycol (PEG400, 0.81 mM), sodium methanesulfonate (NaMS, 1.73 mM) and tetramethylammonium chloride (TMAC, 1.37 mM) in place of Tween 20, SDS and CTAB, respectively. The experimental results, reported in Figure 10, clearly show that, in complete analogy with the results obtained in the presence of surfactants, NaMS and TMAC significantly reduce the binding affinity, while PEG400 exerts a much more limited effect. This confirms the aforementioned hypothesis of an ion pair effect due to the charged head of the surfactants.

## 5. Conclusions

The experimental results indicate that surfactants are able to interfere with the binding between a molecularly imprinted polymer and its ligands, but that this effect is not attributable to any amphipathic properties of these kind of additives, and instead is limited to the ability to form ion pairs with the polymer and the ligand. Consequently, it appears to be impractical to use such surfactants to reduce the non-specific binding without also affecting the molecular recognition properties of the imprinted polymer. 

## Data Availability

Not applicable.

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
