# Peer review of "Effect of Surfactants on the Binding Properties of a Molecularly Imprinted Polymer"

_polymers, 2022, doi:10.3390/polym14235210_

Round 1

Reviewer 1 Report

The manuscript(Manuscript ID: polymers-2036133) authors investigated the effect of three surfactants (the anionic sodium dodecylsulphate, SDS; the cationic cetyltrimethylammonium bromide (CTAB), the non-ionic polyoxyethylene-(20)-sorbitan monolaurate, Tween 20) on the binding affinity of a 2,4,5- trichlorophenoxyacetic acid (2,4,5-T)-imprinted polymer for the template and its analogue 2,4- dichlorophenoxyacetic acid (2,4-D). The experimental results indicated that it appears to be impractical to use such surfactants to reduce the non-specific binding without affecting the molecular recognition properties of the imprinted polymer.

 However, before the manuscript accepted for publication in the journal of POLYMERS, the following minor or major issues should be carefully addressed.

1.      In all the figures, there are no statistical difference indication between every bars. If the authors provided the statistical analysis, it will make the manuscript more readable.

2.       In the Scheme 1, the authors depict the possible dynamic interactions between surfactants, phenoxyacids and phenoxyacid-imprinted polymer. If the authors also show the interaction between the template, functional monomer and surfactant, such as hydrogen bounds together with hydrophobic interactions, it could make the hypothesis seems very plausible.

Author Response

The manuscript (Manuscript ID: polymers-2036133) authors investigated the effect of three surfactants (the anionic sodium dodecylsulphate, SDS; the cationic cetyltrimethylammonium bromide (CTAB), the non-ionic polyoxyethylene-(20)-sorbitan monolaurate, Tween 20) on the binding affinity of a 2,4,5- trichlorophenoxyacetic acid (2,4,5-T)-imprinted polymer for the template and its analogue 2,4- dichlorophenoxyacetic acid (2,4-D). The experimental results indicated that it appears to be impractical to use such surfactants to reduce the non-specific binding without affecting the molecular recognition properties of the imprinted polymer.

However, before the manuscript accepted for publication in the journal of POLYMERS, the following minor or major issues should be carefully addressed.

  1. In all the figures, there are no statistical difference indication between every bars. If the authors provided the statistical analysis, it will make the manuscript more readable.

ANSWER: as correctly pointed out by the reviewer, we have intentionally omitted the error bars from the bar graphs. This decision is essentially due to the fact that even in the case of error bars corresponding to 3 times the standard error, these are difficult to see due to their small numerical value. In reviewing the article we therefore decided to add a support file with the Keq values calculated by non-linear regressions of the binding isotherms

  1. In the Scheme 1, the authors depict the possible dynamic interactions between surfactants, phenoxyacids and phenoxyacid-imprinted polymer. If the authors also show the interaction between the template, functional monomer and surfactant, such as hydrogen bounds together with hydrophobic interactions, it could make the hypothesis seems very plausible.

ANSWER: scheme 1 has been redesigned by adding structural formulas for 2,4,5-T instead of generic symbols

Reviewer 2 Report

1.       Identify equations with number.

2.       The equation of Langmuir binding isotherm model is repeated.

3.       The presentation of Keq results in bar graphs makes it difficult to appreciate trends. It is advisable to modify the way the results are presented to make it easier to appreciate the effect of the two factors analyzed (surfactant concentration and water:acetonitrile mole fraction).

4.       The objective of the present study is not very clear: what is to be demonstrated? The authors' conclusion is that it is impractical to use surfactants to reduce non-specific binding, but it is not very clear how a reduction in non-specific interactions alone could have been proven.

Author Response

  1. Identify equations with number.

ANSWER: Done

  1. The equation of Langmuir binding isotherm model is repeated.

ANSWER: Deleted

  1. The presentation of Keq results in bar graphs makes it difficult to appreciate trends. It is advisable to modify the way the results are presented to make it easier to appreciate the effect of the two factors analyzed (surfactant concentration and water:acetonitrile mole fraction).

ANSWER: we are fully aware that the graphs present some difficulty in being interpreted correctly without reading the text of the article thoroughly and carefully. in fact, while writing the paper, we tried to graphically represent the results in different ways, including dot-line graphs and three-dimensional graphs. The clustered bar graphs that we finally decided to use are those that, in our opinion, allow the best visual interpretation of the experimental results

  1. The objective of the present study is not very clear: what is to be demonstrated? The authors' conclusion is that it is impractical to use surfactants to reduce non-specific binding, but it is not very clear how a reduction in non-specific interactions alone could have been proven.

ANSWER: we are quite surprised that the reviewer did not fully grasp the scope of the work, as in the introduction we explicitly state "With the aim to evaluate the effect of surfactants onto the binding properties of MIPs, in this work we studied the effect of sub-micellar amounts of three different surfactants..." . It follows that we wanted to verify what could be the effect of different surfactants on the interaction between a MIP and its ligand. Anyway, the decrease of the binding affinity for 2,4,5-T on NIP is a acceptable estimate of the effect of surfactants on the non specific binding as clearly stated in the paper

Reviewer 3 Report

In this work, the authors present a study of the effect of three surfactants (the anionic sodium dodecylsulphate, SDS; the cationic cetyltrimethylammonium bromide (CTAB), the non-ionic polyoxyethylene-(20)-sorbitan monolaurate, Tween 20) on the binding affinity of a 2,4,5- trichlorophenoxyacetic acid (2,4,5-T)-imprinted polymer for the template and its analogue 2,4- dichlorophenoxyacetic acid (2,4-D). This is an interesting work and the results were well supported by the experiments. I recommend accepting the manuscript for publication after some details are revised and improved.

1.      How did the effect of three surfactants (SDS, CTAB and Tween 20) on the binding affinity for the template in the presence of 2,4,5-T and 2,4-D?

2.      In the introduction, some typical papers (Nanomaterials, 2022, 12, 2997; Talanta, 2021, 231, 122339) related to MIPs preparation should be properly cited.

3.      The English need to be polished. Some spelling mistakes and language expressions should be checked carefully during the revision.

Author Response

In this work, the authors present a study of the effect of three surfactants (the anionic sodium dodecylsulphate, SDS; the cationic cetyltrimethylammonium bromide (CTAB), the non-ionic polyoxyethylene-(20)-sorbitan monolaurate, Tween 20) on the binding affinity of a 2,4,5- trichlorophenoxyacetic acid (2,4,5-T)-imprinted polymer for the template and its analogue 2,4- dichlorophenoxyacetic acid (2,4-D). This is an interesting work and the results were well supported by the experiments. I recommend accepting the manuscript for publication after some details are revised and improved.

  1. How did the effect of three surfactants (SDS, CTAB and Tween 20) on the binding affinity for the template in the presence of 2,4,5-T and 2,4-D?

ANSWER: the question is not clear. Maybe a verb was omitted. In any case, the whole paper concerns about the effect of surfactants onto the MIP-ligand interactions, so …

  1. In the introduction, some typical papers (Nanomaterials, 2022, 12, 2997; Talanta, 2021, 231, 122339) related to MIPs preparation should be properly cited.

ANSWER: we are sorry to point out that the two papers suggested by the reviewer have no relevance either with the specific topic covered by our work (effect of surfactants onto the MIP-ligand interaction), or in general with the methods preparation of MIPs. We must also observe that the two works in question (but it is certainly a coincidence!) were signed by the same authors. Therefore, it does not seem appropriate or correct to include these works in the bibliography

  1. The English need to be polished. Some spelling mistakes and language expressions should be checked carefully during the revision.

ANSWER: English has been accurately revised

Round 2

Reviewer 2 Report

The authors have clearly answered the questions. The paper can be published in this corrected version.

Author Response

no further changes were requested by the reviewer